# Effects of Hydroxytyrosol in Endothelial Functioning: A Comprehensive Review

**DOI:** 10.3390/molecules28041861

**Published:** 2023-02-16

**Authors:** Ubashini Vijakumaran, Janushaa Shanmugam, Jun Wei Heng, Siti Sarah Azman, Muhammad Dain Yazid, Nur Atiqah Haizum Abdullah, Nadiah Sulaiman

**Affiliations:** Centre for Tissue Engineering & Regenerative Medicine, Faculty of Medicine, Universiti Kebangsaan Malaysia, Jalan Yaacob Latif, Cheras, Kuala Lumpur 56000, Malaysia

**Keywords:** olive, hydroxytyrosol, antioxidant, endothelial functioning, endothelial cells, cardiovascular diseases

## Abstract

Pharmacologists have been emphasizing and applying plant and herbal-based treatments in vascular diseases for decades now. Olives, for example, are a traditional symbol of the Mediterranean diet. Hydroxytyrosol is an olive-derived compound known for its antioxidant and cardioprotective effects. Acknowledging the merit of antioxidants in maintaining endothelial function warrants the application of hydroxytyrosol in endothelial dysfunction salvage and recovery. Endothelial dysfunction (ED) is an impairment of endothelial cells that adversely affects vascular homeostasis. Disturbance in endothelial functioning is a known precursor for atherosclerosis and, subsequently, coronary and peripheral artery disease. However, the effects of hydroxytyrosol on endothelial functioning were not extensively studied, limiting its value either as a nutraceutical supplement or in clinical trials. The action of hydroxytyrosol in endothelial functioning at a cellular and molecular level is gathered and summarized in this review. The favorable effects of hydroxytyrosol in the improvement of endothelial functioning from in vitro and in vivo studies were scrutinized. We conclude that hydroxytyrosol is capable to counteract oxidative stress, inflammation, vascular aging, and arterial stiffness; thus, it is beneficial to preserve endothelial function both in vitro and in vivo. Although not specifically for endothelial dysfunction, hydroxytyrosol safety and efficacy had been demonstrated via in vivo and clinical trials for cardiovascular-related studies.

## 1. Introduction

The endothelium is a single-cell layer that forms the inner cell linings of blood vessels and the lymphatic system. It acts as a semipermeable barrier in exchanging nutrients, fluids, and toxins between tissues and the blood [1]. The endothelium covers about 4000 to 7000 m^2^ and weighs almost 1 kg in the human body [2]. Endothelial cells (ECs) are polarized cells and appear thin, slightly elongated in shape, approximately 30–50 µm in length, 10–30 µm wide, and with a thickness of 0.1–10 µm [1]. This single monolayer structure is responsible for vascular homeostasis, trafficking a wide-range of biomolecules that regulates vascular tone, smooth muscle cell proliferation [3], cell adhesion and thrombosis [4], and inflammation [5]. 

Endothelial cells work via the help of membrane-bound receptors for numerous molecules, including lipid-transporting particles, metabolites, proteins, and hormones, along with specific junction proteins and receptors that direct cell-to-cell and cell-to-matrix interactions [6]. Quiescent ECs have an anticoagulant and non-thrombogenic luminal surface that prevents the adherence of platelets and leukocytes, which inactivate the coagulation mechanism [7]. However, the production of macromolecules by endothelial cells is highly thrombogenic, and activated endothelial cells encourage the thrombus formation that takes place at the basal lamina. Hence, ECs control the balance between thrombosis, hemostasis, and thrombo-resistance [4]. The importance of endothelial cells is better visualized during the wound-healing process. The proliferative phase relies on the endothelial cells enabling tissue growth and survival by forming new blood vessels [8]. 

The endothelial cell (ECs) defect that alters regular vascular homeostasis is often referred to as endothelial dysfunction (ED) [9]. In 1998, Hunt and Jurd set the essential criteria (Figure 1) that define endothelial dysfunction, i.e., (i) upregulation of adhesion molecules, (ii) loss of vascular tone, (iii) phenotype changes, (iv) generation of cytokines, and (v) higher human leukocyte antigen molecules expression [10]. Alexander et al. had recently refined multi-spectrum phenotypes involved in endothelial dysfunction, such as inflammation, de-differentiation, loss of vascular integrity, and permeability [11]. Dysfunctional endothelium decreases nitric oxide (NO) production and subsequently disrupts the vasorelaxation [12]. ECs are also involved in the pathogenesis of atherosclerosis by secreting pro-inflammatory cytokines and platelet-adhesion molecules which reduce vascular wall permeability and thus, permit entry of oxidized lipoproteins and inflammation mediators [13,14,15].

Endothelial dysfunction was found to correlate with cardiovascular diseases [16], hypertriglyceridemia [17], and leukemia [18] in elderly patients. The prevalence and extent of endothelial dysfunction are only partially established. However, a multi-center study showed that endothelial dysfunction (ED) was apparent in 75% of patients who underwent percutaneous coronary intervention (PCI). Endothelial inflammation following surgical revascularisation, i.e., PCI or coronary bypass graft surgery (CABG), increases ED which leads to restenosis [19]. In addition, endothelial dysfunction is also associated with the pathophysiology of hypertension [20,21], chronic kidney failure [22], tumor growth [23], severe viral infectious diseases [24], insulin resistance, and diabetes [5]. Therefore, improving or maintaining endothelial functioning is essential in sustaining homeostasis. 

All major diseases linked to endothelial dysfunction are either closely associated with or will eventually trigger inflammation. Controling reactive oxygen species is a potential approach to circumvent the inflammation (ROS) [25]. The positive association between antioxidants and their effect on ROS has been established. As a good example, ascorbic acid (vitamin C), a potent antioxidant, could mitigate exercise-induced tissue injury if taken as a supplement before exercising [26]. Therefore, in the hope of finding a powerful antioxidant in maintaining endothelial dysfunction, we preselected hydroxytyrosol, which is getting traction as a natural antioxidant supplement and analyzed its correlation, mechanism, and potential in maintaining ED. 

Olives, botanically known as *Olea europaea,* are traditionally found in the Mediterranean [27]. Olive is a tree species with a history of 6000 years, as recorded by the International Olive Council (IOC). Around six million trees are cultivated around Mediterranean countries such as Spain and Greece [28] and further east to Islamic countries like Iran, Palestine, and Iraq [29]. Around 25 kg of olive leaves are obtained annually from pruning each tree [30]. Olive leaves contain secondary metabolites such as oleuropein, oleacein, and hydroxytyrosol [31,32]. Historically, olive leaf is one of the famous herbal teas in the Mediterranean and is a popular prophylaxis as well [33]. Olive phytochemicals were known to exhibit an antioxidant [34,35], antiviral [36], antiproliferative [37,38] and help in cholesterol reduction [39]. In addition, table olives are a well-accepted fermented food in the Mediterranean Diet either as an appetizer or incorporated into recipes such as salads, pizza, and even as a tasty garnish. The international olive council reported that the production of table olives gradually increased from 950,000 to 2,889,000 tonnes from 1990 to 2016 [40]. Table olives are mainly monounsaturated fats with phenolic acids, phenolic alcohols, secoiridoid, and flavonoids [41]. Hydroxytyrosol and tyrosol are the highest phenolic components in table olives, around 76.73 and 19.48 mg/100 g of olives, respectively [42].

The main component in olives is fatty acids, especially monounsaturated fatty acids (MUFA), which contribute up to 85% of an olive’s mass. The rest of the olive contents are chemical compounds such as aliphatic alcohols, triterpene alcohols, sterols, hydrocarbons, pigments, volatile compounds, and phenolic compounds, accounting for 2% of the weight [43]. Olive oil is considered as a traditional symbol of the Mediterranean Diet as it is the main source of fat for Mediterranean people [44]. Olive oil is predominantly composed of monounsaturated oleic acid which is made up of 83% triacylglycerols. Olive oil is enriched with phenolic compounds such as flavonoids, lignans, phenolic acids, and phenolic alcohol [45]. The polyphenol composition in olive oil ranges from 50 to 1000 mg/kg [32] but is less saturated in extra virgin olive oil (EVOO) with 50 to 800 mg/kg [46]. However, the quantity of phenolic compounds in EVOO highly depends on the olive’s maturation, the country’s origin, climate, storage, extraction process, and so on [47]. 

Hydroxytyrosol is the most powerful antioxidant found in olives and olive oil. It is alternatively known as 3-hydroxytyrosol, 3,4-hydroxyphenyl ethanol (DOPET), dihydroxy phenyl ethanol,2-(3,4, dihydroxy phenyl)-ethanol, or 3,4-dihydroxyphenolethanol. This phenolic compound exists as amphipathic in nature with 154.16 g/mol with the chemical formula C_8_H_10_O_3_ [48]. Hydroxytyrosol is extracted during olive maturation and processing of table olive. Olive leaves have the highest concentration of HT [49]. Hydroxytyrosol makes up 40% of total phenolic compounds in olive oil [50]. Meanwhile, the HT content in EVOO ranges from <2 mg/kg up to 14 mg/kg [51] and is estimated to be 20 to 84 mg/l in olive oil [52]. HT has no genotoxicity in humans [46] or animals [53]. The colon and small bowel passively absorb HT. HT absorption rates vary according to the vehicle used, where olive oil was proven to be more efficiently absorbed [54]. Around 98% of HT administered was metabolized into sulphate conjugates or glucuronides, and only 2% was found free in urine and plasma [55]. Hydroxytyrosol reaches maximum plasma concentration in seven minutes and has a 1 to 2 min half-life [56]. Furthermore, a randomized human trial revealed that HT from virgin olive oil becomes a part of high-density lipoproteins (HDL) more rapidly and exhibits antioxidant and cardioprotective properties [57].

Hydroxytyrosol is the only phytochemical with EFSA-approved health benefits (EFSA, 2011) with safety approval [58]. European Food Safety Authority (EFSA) also analyzed free hydroxytyrosol daily intake from olive and table olives from all age groups as shown in Table 1. In the Spanish PREDIMED trial (Prevención con Dieta Mediterránea), a 5-year intervention of the Mediterranean Diet was conducted on 7447 CVD high-risk participants. The Mediterranean Diet significantly decreased clinical manifestations of CVD, including stroke. However, the authors recently retracted the reports and republished them with new analyses with individual randomization [59]. A critical review by Martínez et al. 2019 suggested that a traditional Mediterranean Diet provides better cardiovascular health from the consistence and strong evidence of the original cohorts and randomized controlled trials [60]. 

Hydroxytyrosol (HT) has been studied expansively over the decades [61] for its anti-inflammatory [62,63,64], anti-thrombotic [65], anti-atherogenic [66], hypoglycemic effects [46], and neuro-regeneration [67,68,69]. The antioxidant nature of HT is mainly attributed to the aforementioned health benefits. As a guideline, the European Food Safety Authority (EFSA) panel suggested that a minimum of 5 mg of HT should be consumed to achieve physiological benefits [70]. Consequently, HT began to get more attention from pharmaceutical companies, and HT preparation were patented and marketed as a health supplement. HIDROX^®^ is an example of an olive extract comprised of 40%–50% HT developed and patented by Roberto Crea, PhD, president of CreAgri, Hayward, U.S. (Patent No. 6,165,475 and 6,197,308). 

Furthermore, a recent study by Carluccio et al. [71] reported the nutrigenomic effect of HT in vascular endothelial cells through transcriptomic analysis. Supplementation of HUVEC with HT revealed modulation of inflammatory response that acts via NF-κB signaling. Carluccio et al. [71] further prove various other novel pathways regulated by HT at basal and in an inflamed condition that is crucial in understanding and utilizing HT in circumventing endothelial dysfunction. Although transcriptomic evidence are strong, experimental findings are still lacking to attest to the Ingenuity Pathway Analysis (IPA). Therefore, an apparent gap of HT action at the molecular level, bioavailability, ideal dosages, and clinical data availability are noticed. Data from randomized control trials show that olive oil supplementation reduced endothelial inflammation markers [72], but the effect of HT on endothelial functioning has not been validated through human trials. Accordingly, we collate recent findings of HT effects focusing on endothelial functioning and analyze published in vitro and in vivo studies that reported on HT effects on endothelial functioning in terms of oxidative stress markers, cellular proliferation, migration, and inflammatory biomarkers expression. 

## 2. Hydroxytyrosol in Oxidative Stress-Induced Endothelial Dysfunction

Oxidative stress plays a vital role in endothelial dysfunction. Nitric oxide (NO), also known as endothelium-derived relaxing factor [73], is derived from the oxidation of L-arginine amino acid by nitric oxide synthase [74]. It facilitates vascular homeostasis by maintaining vascular integrity and inhibiting platelet aggregation and neutrophil adhesion [75,76]. Nitric oxide also prevents the proliferation of artery smooth muscle cells in defense against atherosclerosis and neointima hyperplasia formation [77,78]. In addition, an increase in oxidative stress causes superoxide anion to degrade NO and form peroxynitrite ONOO− [79] which decreases the bioavailability of NO while triggering endothelial impairment [73]. On the other hand, reactive oxygen species (ROS) is a fundamental aspect of oxidative stress in endothelial cells (ECs). Hydrogen peroxide, superoxide anions, and hydroxyl radicals are the sources of the intracellular ROS production [80]. Zheng et al. [81] systematically reviewed the sources of endothelial ROS production and their underlying mechanism on ECs death. Moreover, ROS also increases intracellular calcium concentration and activates proto-oncogenes and pro-inflammatory genes [82]. Ironically, antioxidants promote cellular health by scavenging ROS. Silva et al. [83] elaborate in detail that decreased NO availability leads to endothelial dysfunction and disrupts vascular tone regulation. Oxidative stress in ED is observed in the clinical setting with renovascular hypertension and Gilbert syndrome patients, which suggest ED-related oxidative stress could be a therapeutic target for atherosclerosis [84]. Hydroxytyrosol has been proven to reduce oxidative markers such as ROS and malondialdehyde (MDA) and upregulate NO generation in vitro and in vivo [62,85,86,87]. The ortodiphenolic structure of HT is responsible for scavenging free radicals [28]. Serreli et al. [88] reported conjugated metabolites of hydroxytyrosol and tyrosol-enhanced endothelial nitric oxide synthase (eNOS) expression and decreased superoxide production by activating AKT serine/threonine kinase 1 (AKT 1). However, two studies reported contradictory outcomes when HT failed to increase NO production [82,83] significantly. Interestingly, both studies utilized the exact same dosage of 30 μM HT. In contrast, studies that utilize higher HT concentrations between 50 to 100 μM, reported a significant NO generation. Therefore, a higher HT dose between 50 to 100 μM could efficiently promote NO production in vitro. Another key player in oxidative stress is sirtuin 1 (SIRT1), a part of the seven-protein family that regulates cellular activity [89]. SIRT1 regulate vascular NO generation and vascular aging. Impaired SIRT1 causes oxidative stress and inflammation [83] which accelerates the vascular aging [90]. Hydroxytyrosol-nitric oxide (HT-NO) compound reduced ROS generation by activating sirtuin 1 (SIRT1) [91]. Bayram et al. [92] reported that olive oil enriched with HT supplementation had reduced oxidative biomarkers in senescence-accelerated mouse-prone 8 (SAMP8) mice, which were observed via upregulation of SIRT1 mRNA expression and the nuclear factor erythroid 2–related factor 2 (Nrf2)-dependent gene expression. Hydroxytyrosol was reported to upregulate SIRT1 to salvage TNF-α-induced vascular adventitial fibroblasts [93]. Thus, HT primarily targets the SIRT1 gene and protein activation to maintain endothelial functioning. 

Nuclear factor-E2-related factor 2 (Nrf2) is a transcription factor that binds to antioxidant response elements that regulate numerous antioxidant genes [94]. Nrf2 is one of the most potent antioxidant pathways potentially preventing oxidative injuries in vascular endothelial cells [95,96]. Hydroxytyrosol at 50μM increases the nuclear accumulation of Nrf2 and heme-oxygenase 1 (HO-1) expression while mediating wound healing [97]. The bioactivity is mediated by the phosphatidylinositol 3-kinase/ AKT serine/threonine kinase 1 (PI3K/Akt) and extracellular signal-regulated kinase1/2 (ERK1/2) signaling pathways as visualized in Figure 2. However, no difference in NO production was observed. A new derivative of hydroxytyrosol, peracetylated hydroxytyrosol (Per-HTy) showed anti-inflammatory activity by increasing Nrf2 and HO-1 in lipopolysaccharide-induced murine macrophages [98]. Apart from hydroxytyrosol, Parzonko et al. tested oleuropein and oleacein on endothelial progenitor cells, successfully preventing cell damage induced by angiotensin II by inducing Nrf2-HO-1 expression. HT and oleic acid were also shown to increase HO-1 expression in murine dermal fibroblast [99]. Consequently, HT protects against oxidative stress and potentially increases the re-endothelialisation ability of injured arterial walls and neovascularisation of the ischemic tissue [100]. HT also protected endothelial cells from H_2_O_2_ by increasing Nrf2 expression and activating Akt and ERK1/2 [101]. Furthermore, olive oil has also been shown to activate AKT phosphorylation and Nrf2 expression while mediating migration and proliferation in dermal fibroblast [102]. 

Sindona et al. [103] reported that 3,4-DHPEA-EDA, a dialdehyde oleanolic acid linked to hydroxytyrosol and 3,4-DHPEA-EA oleuropein aglycone, enriched NO generation, while HT alone does not offer the same results. In contrast, 50 μM HT upregulated NO production by activating eNOS protein expression in porcine artery-derived endothelial cells [87]; whereas 100 μM of HT and hydroxytyrosol acetate (HC) suppresses the production of oxidative stress markers, malondialdehyde (MDA) and ROS, thus increasing the production of superoxide dismutase (SOD) in TNF-induced HUVECs [104]. Hydroxytyrosol at 30 μM reduces phorbol myristate acetate (PMA)-induced mitochondrial superoxide production in human umbilical vein endothelial cells (HUVECs) [105]. Furthermore, HT has been a mitochondrial ROS scavenger in the PMA-inflamed HUVECs [106]. HT also reduces mitochondrial superoxide production by enhancing the superoxide dismutase SOD activity [105]. The inconsistencies and contrasting outcomes could be a limitation of hydroxytyrosol application in clinical trials.

The bioavailability and half-life of hydroxytyrosol are crucial in validating HT effects in general. Hydroxytyrosol undergoes phase-I and phase-II metabolism in the intestine and liver, leading to poor bioavailability [107]. Oleuropein, oleacein, and ole aglycone are hydrolyzed in the digestion system and generate HT as metabolites [108]. Thus, research on the activity of HT metabolites is equally significant. Hydroxytyrosol and its glucuronide metabolites (GC) decrease superoxide production in rat aortic ring experimental set-up [109]. Generally, 1 μM of HT is sufficient to provide minimal activation of antioxidant activity [110,111,112,113]. Besides, a range between 1 to 100 μM of HT [87,105,110,111,113,114] is typically utilized for in vitro studies. Vissers et al. [115] reviewed the metabolism, bioavailability, and excretion of olive oil phenols. They concluded that a range of 50 up to 100 μM is required to exhibit an antioxidant effect in vitro. A summary of biomolecules and molecular pathways regulated by hydroxytyrosol in endothelial functioning is illustrated in Figure 2. 

Concerning in vivo studies, 5 to 80 mg/kg of hydroxytyrosol was reported to exhibit positive health benefits [111,114,116]. Catalan et al. [111] reported a significant increase in HT phase-II metabolites such as HT sulphate and homovanillic acid sulfate in urine and feces, post 24 h of olive extract supplementation. Moreover, supplementation of 10 mg/kg olive extract to mice showed a significant decrease in E-selectin, MCP-1, and ICAM-1 expression in tunica intima, media, and adventitia region [111]. However, no significant differences were observed in lipid profile and atherosclerotic lesions. Interestingly, 50 mg/kg of hydroxytyrosol and its derivatives significantly reduced lipid peroxide and increased GSH [114]. HT also inhibited thromboxane synthesis, and weakly lowers prostacyclin production which demonstrates their anti-platelet aggregation effects [114]. However, the best dosage could not be determined due to high variability in the dose range, duration of treatment, frequency of treatments, and cell types utilized in all studies with HT. Rest assured that the concentration mentioned earlier, or dose ranges, do not show any adverse effect in all in vitro and in vivo studies. Studies conducted on the hydroxytyrosol effect in oxidative stress are listed in Table 2.

## 3. Hydroxytyrosol Defence in Endothelial Dysfunction-Induced Atherosclerosis

Vascular inflammation is a precursor for endothelial dysfunction (ED). ED recruits platelets and immune cells, triggering the production of inflammatory cytokines and chemokines to subdue inflammation [123]. However, persistently elevated generation of these inflammatory molecules is detrimental to vascular cells. Dysfunctional ECs are activated; thus, expressing adhesion molecules such as vascular cell adhesion molecule (VCAM-1) and intercellular adhesion molecule 1 (ICAM-1), which in turn start to recruit inflammatory cells contributing to the progression of the atherosclerosis [124]. Hydroxytyrosol suppresses the production of E-selectin, P-selectin, VCAM-1, and ICAM-1 in TNF α-induced porcine aortic ECs proving HT mechanism in managing inflammation [110]. HT at 1 to 30 μmol/L reduces PMA-induced TNF-α, IL- 1β, VCAM-1, and ICAM-1 production in HUVECs [105]. Interestingly, HT plasma metabolite, hydroxytyrosol-3-O-sulfate, reduces adhesion molecules, including monocyte chemoattractant protein-1(MCP-1), whereas HT on its own failed to exert the same effect [110]. 5 µM hydroxytyrosol-3-O-sulfate effectively reduces E-selectin and VCAM-1, even though HT as is failed to [110]. In another study, HT impedes cardiovascular biomarkers platelet aggregation, VCAM-1, and thromboxane B_2_ [125]. 

Hydroxytyrosol acetate exerts an anti-inflammatory effect via the SIRT6-mediated PKM2 signaling pathway and inhibits TNF activity by TNF receptor superfamily member 1A (TNFRSF1A) signaling pathway [104]. Activation of SIRT6 improved endothelial function in haplo-insufficient mice by inhibiting NAD(P)H oxidase, which is responsible for oxidative stress [126]. Hydroxytyrosol also countered systemic inflammation in mice by inhibiting the expression of inflammatory markers, cyclooxygenase-2 (COX2), IL-1b, IL-6, and TNF-α via mediating the SIRT6/PKM2 signaling pathway [85]. SIRT6 regulates autophagy in HUVECs and diabetic-induced mice as well [127]. Similarly, SIRT6 overexpression decreased atherosclerotic lesions and ECs dysfunction markers in both mice and humans [128]. Hence, HT can potentially manage ED indirectly by mediating the SIRT6 pathway in addition to regulating oxidative stress via inhibiting pro-inflammatory molecules. 

Foam cell formation in atherosclerosis also decreased with hydroxytyrosol supplementation. Macrophages tend to penetrate damaged endothelial lining and subsequently transform into foam cells by engulfing the oxidized LDL [129]. Hydroxytyrosol reduces foam cell formation and flavin-containing monooxygenase 3 (FMO3), a cholesterol efflux modulator expression induced by acrolein [112]. Additionally, HT regulates the reverse cholesterol transport pathway by increasing ATP-binding cassette transporter (ABCA1) expression. ABCA1 is responsible for the efflux of cholesterol from lipid-laden macrophages. Enhanced ABCA1 protein levels potentially increased cholesterol efflux [130,131]. HT has also been shown to lower FMO3 and lipid recruitment while increasing ABCA1 expression in the acrolein-induced atherogenesis [112]. The aggregation of lipids in the endothelial space is a subsequent event of ED that leads to atherosclerosis. In the development of atherosclerosis, impaired cholesterol and low-density lipoprotein (LDL) transport enable LDL to infiltrate into subendothelial space, oxidized, and subsequently taken up by macrophages [132]. HT were found to activate the AMPK pathway and phosphorylation of p38 which subsequently regulates lipid metabolism in atherosclerotic mice [133]. Pathways affected by hydroxytyrosol are depicted in Figure 2. A systematic review and meta-analysis conducted by Pastor et al. [134] reported HT dampers LDL lipid production, which is beneficial in preventing atherosclerosis but the findings were not supported by a 95% confidence interval. 

## 4. Hydroxytyrosol Prevents Endothelial-to-Mesenchymal Transition

Endothelial-to-mesenchymal transition (EndMT) is a state of acquiring molecular and cellular changes into mesenchymal-like phenotype by endothelial cells [135,136]. EndMT leads to modification of cell proliferation, contraction, migration potential, and loss of cell polarity [137]. These pathological changes give rise to cardiovascular complications like neointima hyperplasia [138,139] and atherosclerosis [140,141,142]. Terzuoli et al. [143] demonstrates that hydroxytyrosol sulfate metabolite HT-3Os deplete IL-1β-induced EndMT markers like notch receptor 3 (NOTCH3), while matrix metalloproteinase-2 (MMP2) and matrix metalloproteinase-9 (MMP9) reversed the process by upregulating let-7 miRNA expression, CD31, and downregulating the transforming growth factor beta 1 (TGF-β) signaling pathway (Figure 2) [143,144]. Similarly, olive extract prevented TGFβ1-induced airway epithelial-to-mesenchymal transition by reducing vimentin expression and reversing its native morphology [145]. Inflammation and altered shear-stress drives are among a few of the EndMT manifestations that enhance the atherosclerosis [136]. Evrard et al. [146] studied single-cell imaging of mesenchymal cells transitioned from endothelium-in-atherosclerotic aortas where 3 to 9% of the intimal plaque cells are found to be of ECs origin. EndMT-derived smooth muscle cells via uncontrolled TGFβ generation were also found to aggravate intimal thickening in newly grafted vessels [147]. We have previously summarized hydroxytyrosol’s potential and molecular action in the re-endothelialisation and prevention of the intimal thickening [148]. Hydroxytyrosol has the potential to attenuate intimal thickening by reversing EndMT. 

## 5. Hydroxytyrosol Ameliorates Angiogenesis and Wound Healing

Tube formation and migration are essential in angiogenesis, wound healing, and tissue regeneration of injured tissues. In this context, by ingesting olive oil, hydroxytyrosol accelerates cell migration and angiogenesis at low doses of 1 to 5 µM [113]. HT also promotes wound healing and ECs capability by upregulating matrix metalloproteinase-2 MMP-2, Rho/Rho-associated coiled-coil containing protein kinase (Rho/ROCK), Phospo Src, Phospho Erk1/2, Erk1/2, RhoA, Rac1, and Ras protein expressions [113]. Furthermore, HT accelerates vascular formation with concurrent overexpression of vascular endothelial growth factor receptor VEGF-R2 and activation of the PI3K-Akt-eNOS signaling pathway. Hydroxytyrosol was also found to promote angiogenic through HIF-1 [149]. Co-treatment of hydroxytyrosol with pulsed electromagnetic fields (PEMFs) on HUVEC increases cell proliferation, migration, and protection against H_2_O_2_-induced apoptosis via upregulation of Akt, mTOR, and TGF-β expression [150]. Conversely, HT facilitates wound healing in porcine pulmonary artery-derived ECs via inducing HO-1 and Nrf2 mRNA expression and PI3K/Akt /ERK1/2 pathways [97]. 

## 6. Epigenetic Effect of Hydroxytyrosol in Endothelial Functioning

The epigenetic mechanism is defined as gene regulation that includes DNA methylation, microRNA (miRNA), and histone modification that alters phenotypic changes without changing the DNA sequence [151,152]. Polyphenol treatment that induces epigenetic mechanism is referred to as an “epigenetic diet [153].” DNA methylation is a part of the epigenetic mechanism where a methyl group is transferred onto the C5 position of the cytosine to form 5-methylcytosine to repress gene expression [154]. Hydroxytyrosol treatment reversed H_2_O_2_-induced miR-9 promoter’s hypomethylation during oxidative stress conditions [155]. Lopez et al. [156] analyzed transcriptomics of HT-supplemented mice and found two novel targets of HT fibroblast growth factor 21 (Fgf21) and a liver-secreted peptide hormone RORA (RAR Related Orphan Receptor A) in lipid metabolism. Recently, a systematic review was conducted on the nutrigenomic effects of phenolic compounds of extra virgin olive oil (EVOO) and olive oil involving in vitro, in vivo, and human studies [157]. They concluded EVOO and olive oil polyphenols are able to modulate epigenetic mechanisms. Hydroxytyrosol was found to regulate miRNAs in cardiovascular, cancer, and neurodegenerative diseases.

## 7. The Effects of Hydroxytyrosol and Its Derivatives on Hypertension

Hypertension is a commonly occurring chronic medical condition and is currently defined and characterized by the presence of a persistent elevation in systolic blood pressure (SBP) and/or diastolic blood pressure (DBP) to values above 130 mmHg and 80 mmHg, respectively [158]. It also represents one of the most important modifiable cardiovascular risk factors, with approximately 54% and 47% of all global stroke and coronary artery disease cases due to hypertension [159]. Multiple studies have shown that supplementation with olive polyphenolic compounds such as HT and its derivatives reduce SBP and DBP [160,161]. Clinical trials conducted by Hermans et al., involving a total of 663 patients, recorded a mean SBP reduction of 13 ± 10 mmHg and DBP reduction of 7.1 ± 6.6 mmHg after 8 weeks of supplementation with 100 mg oleuropein (Ole) and 20 mg HT daily, with larger reductions seen in patients with higher baseline blood pressures [161]. Lockyer et al. [162] observed a significant, albeit modest reduction in mean-patient daytime SBP/DBP values of −3.95 ± 11.48 /−3.00 ± 8.54 mmHg, 6 weeks post-treatment with 136 mg Ole, and 6 mg HT compared to their control group. Similarly, a randomized double-blind clinical trial conducted by Quirós-Fernández et al. [160] revealed that supplementation with hydroxytyrosol and punicalagin (PC), another phenolic compound derived from pomegranates, yield a significant −15.75 ± 9.9 mmHg reduction in the SBP of pre-hypertensive and hypertensive patients compared to their control group. The anti-hypertensive effects shown by HT are likely attributed to its reactive oxygen species (ROS)-scavenging capabilities and its modulation of redox-sensitive signaling pathways which affect a combination of factors, among them being enhanced endothelial nitric oxide synthase (eNOS) production [163]. Endothelial dysfunction is characteristic of hypertensive patients due to an increase in the production of the superoxide (O_2_^-^) anion that can readily and quickly bind to NO, producing peroxynitrite (OONO^-^) which renders ECs inefficient [164]. Thus, deficiencies in eNOS and systemic nitrite levels are generally associated with hypertension and other pathophysiological cardiovascular modifications (i.e., atherosclerosis, coronary artery disease) in both animal models and humans given its importance in BP and flow regulation (e.g., vasodilation, inhibition of platelet aggregation, and inhibition of vascular smooth muscle cell growth) [165,166,167].

Supplementation with Ole, which can be enzymatically hydrolyzed into HT, generated a prophylactic effect that prevented the onset of hypertension in rats and mice in a dose-dependent fashion, attributing to the upregulation of nitric oxide synthase protein expression [168,169]. Storniolo et al. [170] investigated the effects of HT in combination with other polyphenolic extracts from extra virgin olive oil using an in vitro model deficient in intracellular NO due to impaired eNOS phosphorylation. HT treatment increased phosphorylated eNOS with the phosphorylation process mediated through the PI3K/Akt signaling pathway, resulting in increased intracellular NO levels. The treatment group also showed a reduction in levels of endothelin-1 (ET-1), a potent vasoconstrictor which acts in contrast to NO and is highly expressed in hypertensive patients [170]. A combination of these factors contributes to the antihypertensive effects of HT and its derivatives [170,171]. 

HT’s effects on NO could result from NAD^+^-dependent deacetylase and sirtuin 1 (SIRT1) activation, which is involved in the regulation of NO production and NO-mediated endothelium-dependent vascular relaxation through the deacetylation and activation of eNOS [172]. Molecular docking studies showed good compatibility between HT and SIRT1, allowing both to directly bind to each other to activate SIRT1-mediated downstream signaling pathways such as the SIRT1-FOXO1-SOD1 pathway which has been shown to exert anti-hypertensive effects [91,93,173]. Increased levels of phosphorylated eNOS were detected post-treatment with a hydroxytyrosol-nitric oxide composite (HT-NO), but eNOS levels were decreased when SIRT1 expression or function was inhibited [91,93]. A separate in vivo study observed increased levels of activated eNOS when SIRT1 expression was upregulated [174], suggesting that certain compounds can indeed regulate eNOS activity and NO production; HT being one of them. 

HT and its derivatives possess calcium antagonistic properties which play a significant role in blood pressure control and, therefore, their anti-hypertensive effects [175]. Gilani et al. showed that the reduction in mean arterial BP seen in olive extract-treated rats was dose-dependent, which is a common characteristic for calcium channel blockers, and followed a similar trend to verapamil, signifying that the anti-hypertensive effect of the treatment is likely calcium channel-mediated [175]. Scheffler et al. [176] obtained comparable results in their study and concluded that Ole suppresses L-type Ca^2+^ channels directly and reversibly in a dose-dependent fashion. These studies suggest that Ca^2+^ antagonism is one of the mechanisms of action for which HT and its derivatives carry out their antihypertensive effects. Hydroxytyrosol and its derivatives demonstrate a positive anti-hypertensive effect, mainly occurring through SIRT1-mediated upregulation of eNOS deacetylation and phosphorylation, resulting in higher systemic NO levels promoting vasodilation and other BP-lowering effects. Their calcium channel-blocking properties reduce vascular resistance and reactivity to vasoconstriction-promoting and BP-raising signals from the sympathetic nervous system and RAAS, among others. 

## 8. Hydroxytyrosol and Vascular Aging-Induced Endothelial Dysfunction

Aging is a continuous process where the body’s physiological function starts to decrease starting from early adulthood. It is an unavoidable risk factor leading to cardiovascular diseases (CVD), as the prevalence of CVD increases with age [177]. In addition, despite being in excellent cardiovascular health condition, one cannot reverse the effect of aging in the progressive decline of cardiovascular homeostasis and function. As people age, the blood vessels undergo physiological changes leading to reduced vascular function, deterioration of endothelial function, and arterial stiffness [178]. There are several mechanisms involved in the progression of age-related endothelial dysfunction. The increase in oxidative stress, vascular inflammation, and shifts in conserved molecular pathways all contribute to the aging process [179]. One hallmark of endothelial dysfunction is the reduction of nitric oxide (NO). NO is a critical vasodilating molecule important for anti-thrombotic, anti-inflammatory and antioxidant functions [180]. Several age-related factors have been shown to contribute to the imbalance in the production and degradation of NO that results in endothelial dysfunction. Studies show increased arginase activity in the elderly. This, in return, leads to a deficiency of L-arginine and a reduction of NO [181]. In addition, the scavenging of NOs by reactive oxygen species during age-induced oxidative stress can reduce NOs availability [179]. Aged-induced oxidative stress produces a high concentration of O^-^_2_ which lead to the inactivation of NO and upregulation of peroxynitrite (ONOO^-^) and nicotinamide adenine dinucleotide phosphate (NADPH), which further induce oxidative stress, continuing this vicious cycle of NO depletion [182]. 

Inflammatory responses are also associated with vascular aging. Chronic low-grade inflammation is a characteristic of age-related diseases which has been demonstrated by an increase in pro-inflammatory molecules in aging animal models [179,183]. The inflammatory responses are initiated by the release of pro-inflammatory cytokines such as tumor necrosis factor-alpha (TNF-α) and interleukin-1 (IL-1β). This resulted in an increased expression of cell adhesion molecules (CAMs) expressed by the endothelial cells resulting in the recruitment and adhesion of circulating leukocytes to the endothelial wall [184]. Thus, changes to the structure of the vessel wall leading to endothelial dysfunction resulted. Both oxidative stress and inflammation reduce NO bioavailability, which can impair endothelial function in the elderly. An animal study by Wang et. al. shows that mice treated with a high dose of HT produce more antioxidant enzymes and anti-inflammatory cytokines [64]. HT exerts its anti-inflammatory property by down-regulating the expression of iNOS; thus, inhibiting the activation of NF-κB [185]. Since HT has been shown to exhibit antioxidant and anti-inflammatory properties, more studies were performed to look at the effect of HT in preventing age-related diseases such as neurodegenerative disorders [186,187]. Many studies have been done to ascertain the effect of HT on neurodegenerative diseases. However, not many have ventured into the impact of HT on vascular aging. As the mechanisms leading to vascular aging are closely related to other age-related diseases with the hallmark of increased oxidative stress and increase systemic inflammation, it is intriguing to see the potential of HT in attenuating vascular aging.

## 9. Conclusions

In this present review, we reported the current findings of HT efficacy in endothelial functioning. We discussed the positive impact of HT on managing oxidative stress, anti-inflammation, EndMT transition, arterial stiffness, and vascular aging. Most of the in vitro studies revealed that a lower concentration of HT can exert vascular benefit, which could be achieved through oral consumption of EVOO. Although EFSA recommends 5 mg of HT daily, HT doses have yet to be tested in a larger population for a more extended period to standardize personalized HT supplementation. More clinical trials on HT supplements or therapy for endothelial dysfunction-related diseases will be beneficial to deepen the understanding of HT efficacy in vascular diseases. This review allows the scientific community to get the general overviews of what has been done and achieve, and what more could be done before moving towards establishing a potential clinical therapy to salvage endothelial dysfunction or a nutraceutical supplementation regime to achieve protection of endothelial function utilizing hydroxytyrosol. 

## Figures and Tables

**Figure 1 molecules-28-01861-f001:**
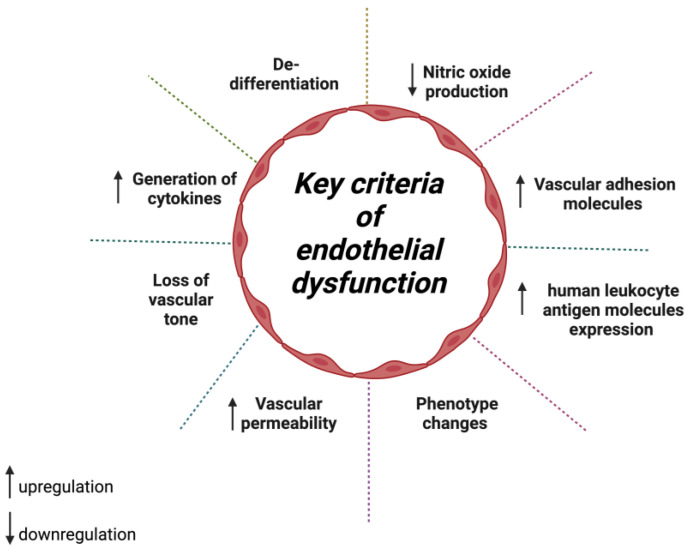
Critical criteria to define endothelial dysfunction.

**Figure 2 molecules-28-01861-f002:**
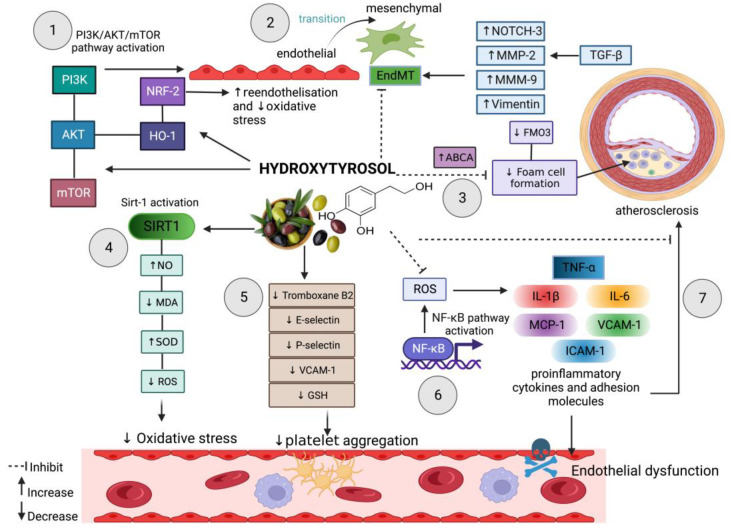
Summary of biomolecules and molecular pathways regulated by hydroxytyrosol in endothelial functioning. (1) HT activates PI3K/AKT/mTOR in re-endothelisation and Nrf2/HO-1 expression in promoting re-endothelisation and decreasing oxidative stress. (2) HT inhibits TGF-β-induced EndMT transition. (3) HT regulates the reverse cholesterol transport pathway by increasing (ABCA1) and decreasing FMO3. (4) HT suppresses oxidative stress by activation of SIRT1. (5) HT lowers platelet aggregation by suppressing adhesion molecules. (6) HT inhibits NF-kβ-induced ROS which subsequently triggers the production of pro-inflammatory cytokines and adhesion molecules. (7) HT inhibits the production of pro-inflammatory cytokines and adhesion molecules that promotes atherosclerosis.

**Table 1 molecules-28-01861-t001:** Free hydroxytyrosol daily intake from olive and table olive based on the EFSA Comprehensive European Food Consumption Database. [58].

Population Group	Free Hydroxytyrosol Daily Intake from the Consumption of Olive Oils		Free Hydroxytyrosol Daily Intake from the Consumption of Table Olives	
	Range of Means (mg/kg bw/day)	Range of 95% (mg/kg bw/day)	Range of Means (mg/kg bw/day)	Range of 95% (mg/kg bw/day)
Children (3–9 years)	0.00015–0.008	0.0003–0.016	0.019–0.375	0.059–0.270
Adolescents (10–17 years)	0.00015–0.005	0.00046–0.008	0.013–0.204	0.059–0.382
Adults (18–64 years)	0.00015–0.004	0.00046–0.007	0.019–0.185	0.079–0.415
Elderly (≥65 years)	0.00007–0.004	0.0016–0.007	0.013–0.125	0.059

**Table 2 molecules-28-01861-t002:** Studies on hydroxytyrosol effects on oxidative stress and anti-inflammatory action.

Oxidative Stress	Inflammation
HT ↓ PMA stimulated mitochondrial superoxide [105]	HT ↓ PMA stimulated pro-inflammatory cytokines (TNF-α, IL- 1β, VCAM-1, ICAM-1) [105]
HT ↑ mitochondrial biogenesis [105]	HT and HT metabolites ↓ inflammatory cytokines (E-selectin, P-selectin, VCAM-1, and ICAM-1) [110,117]
HT ↑ NRF-1 gene expression [97,101,105]	HT ↓ cardiac markers (LDH, creatine kinase, and troponin-T) and TNF-α and IL-6 level in in LPS induced cardiac dysfunction [118]
HT ↑ NO production [87,88,91,118]	HT-acetate lowers SIRT6 protein and mRNA expression [62,85]
HT and HT-AC ↑ SOD and ↓ MDA, ROS [62,118]	Hydroxytyrosol inhibited PMA induced prostaglandin (PG)E2 and COX-2 expression [119]
HT ↑ eNOS, PGC1α by counter reacting with TNF-α [117]	HT ↓ TNF-α induced IκBα and NFκBp65 phosphorylation [87]
HT derivatives ↑ GSH [110,120] ↑ NO [110] ↓ ROS [121]	HT and HT-AC ↓ TNF-α induced IL-1β, IL-6 and CCL2 [62]
HT ↑ HO-1 mRNA, and Nrf2, PI3K/Akt, and ERK1/2 protein expression in [97]	HT metabolites ↓ E-selectin, ICAM-1 and VCAM-1 [111]
HT ↓ H_2_O_2_ induced oxidative stress, apoptosis, and DNA damage [122]	3,4-DHPEA-EDA ↓ LPS- and TNF- α induced E-selectin, ICAM- and VCAM-1, CCL2 and NF-kB activation [103]

Phorbol myristate acetate (PMA); interleukin-1 beta (IL- 1β); vascular cell adhesion molecule (VCAM-1); intercellular adhesion molecule 1 (ICAM-1); nitric oxide (NO); nuclear factor-E2-related factor 2 (Nrf2); reactive oxygen species (ROS); malondialdehyde (MDA); glutathione (GSH); AKT serine/threonine kinase 1 (AKT 1); sirtuin 1 (SIRT1); heme oxygenase 1 (HO-1); lactate dehydrogenase (LDH); prostaglandin E2 (PGE2); tumor necrosis factor α (TNF-α); nuclear factor of kappa light polypeptide gene enhancer in B-cells inhibitor, alpha (IκBα); 3,4-DHPEAEDA, [2-(3,4-hydroxyphenyl) ethyl (3S,4E)-4-formyl-3-(2-oxoethyl)hex-4-enoate; monocyte chemoattractant protein-1(MCP-1), cyclooxygenase-2 (COX2); hydrogen peroxide H_2_O_2_; c-c motif chemokine ligand 2 (CCL2); nuclear factor-κB (NF-kB). ↑ = Increases/upregulate. ↓ = Decreases/downregulate

## Data Availability

Not applicable.

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
