# Peer review of "Effects of Hydroxytyrosol in Endothelial Functioning: A Comprehensive Review"

_molecules, 2023, doi:10.3390/molecules28041861_

Round 1

Reviewer 1 Report

Research on the protective role of hydroxytyrosol (HT) on vascular function is a subject of active research work. This comprehensive review aims to summarize recent data regarding the effects of HT in endothelial functioning at the cellular and molecular levels reported in literature.

The manuscript has several concerns that should be addressed.

The abstract is not very impactful. It should be enhanced by reporting an objective representation of the review.

The authors describe olives and olive leaves as sources of HT in the Mediterranean area, but they did not report that olive oil is the symbol of the Mediterranean diet. Virgin olive oil and extra virgin olive oil are the main sources of active phenolic compounds, responsible for the lower mortality and morbidity observed in people following the Mediterranean diet. Authors should describe the beneficial effects and content of HT and its derivatives in olive oil.

There are too many abbreviations in the text making it difficult to read. These abbreviations are often used incorrectly or the full name is not given. Authors should limit the use of abbreviations and quote the full name before the abbreviation and use the abbreviation afterwards both in the main text and in figures or tables.

Furthermore, in lines 76-78 the authors report: “Therefore, in the hope of finding a powerful antioxidant in maintaining ED, we preselected hydroxytyrosol, which is getting traction as a natural antioxidant supplement and analyzed its correlation, mechanism, and potential in maintaining ED”. Does ED indicate endothelial dysfunction?

The efficacy of HT in vivo needs to be carefully evaluated considering the absorption kinetics and metabolism of this compound after ingestion. The authors should better describe this important topic.

The authors described the effect of HT on the expression of molecules involved in endothelial function. It is known that the expression of these molecules can be regulated by epigenetic mechanisms such as DNA methylation, post-translational histone modification, and non-coding microRNA activity. The authors could discuss the epigenetic effect of HT in regulating the expression of molecules involved in endothelial function.

Figure 1 shows an up arrow for NO production. Please clarify the meaning of the arrow.

Furthermore, Figure 3 lacks a caption describing the effects of HT and the underlying mechanisms of action. Please enter the caption of the figure, the abbreviations and the meaning of the arrows (up and down arrow).

In the conclusion (lines 448-449) the authors report "We strengthened the correlation between HT and oxidative stress, anti-inflammation, EndMT transition, arterial stiffness, and vascular ageing". Please clarify the sentence.

Author Response

Dear Reviewer thank you for your valuable comments. We have tried our best to rectify all the mistakes and added required informations

Reviewer 2 Report

Dear Authors,

In the review intitled "Effects of Hydroxytyrosol in Endothelial Functioning: A Comprehensive Review" you reviewed the current findings of Hydroxytyrosol's efficacy in endothelial functioning, investigated the relationship between HT and oxidative stress, inflammation, the EndMT transition, arterial stiffness, and vascular aging in vitro and in vivo. The review is well-written, but still some thinks need to be changed to improve the quality of the manuscript and I ask please to carefully read. You will find the revisions in the form of comments in the attached manuscript.

Good work

Author Response

(The authors gave the same response as above.)

Round 2

Reviewer 1 Report

The authors took into account the reviewer's comments and adapted the manuscript to improve its overall quality. However, Figure 1 is still misleading. The authors argued that the up arrow represents an increase, but this is incorrect with respect to nitric oxide production. Please correct and add caption to Figure 1.

Author Response

Dear Reviewer, sorry for the mistake. Initially, we replied to the comment based on figure 2. We apologise for that. We have amended the arrow in figure 1 and the caption for the arrows was added. Thank you